# Smooth Normalizing Flows

**Jonas Köhler**[*][†]      **Andreas Krämer**[*][†]      **Frank Noé**[†][‡][§]

[†] Department of Mathematics and Computer Science, Freie Universität Berlin
[‡] Department of Physics, Freie Universität Berlin
[§] Department of Chemistry, Rice University, Houston, TX
{jonas.koehler, andreas.kraemer, frank.noe}@fu-berlin.de

## Abstract

Normalizing flows are a promising tool for modeling probability distributions in physical systems. While state-of-the-art flows accurately approximate distributions and energies, applications in physics additionally require smooth energies to compute forces and higher-order derivatives. Furthermore, such densities are often defined on non-trivial topologies. A recent example are Boltzmann Generators for generating 3D-structures of peptides and small proteins. These generative models leverage the space of internal coordinates (dihedrals, angles, and bonds), which is a product of hypertori and compact intervals. In this work, we introduce a class of smooth mixture transformations working on both compact intervals and hypertori. Mixture transformations employ root-finding methods to invert them in practice, which has so far prevented bi-directional flow training. To this end, we show that parameter gradients and forces of such inverses can be computed from forward evaluations via the inverse function theorem. We demonstrate two advantages of such smooth flows: they allow training by force matching to simulation data and can be used as potentials in molecular dynamics simulations.

## 1 Introduction

Generative learning using *normalizing flows* (NF) [50, 42, 39] has become a widely applicable tool in the physical sciences which has e.g. been used for sampling lattice models [36, 37, 31, 4, 1], approximating the equilibrium density of molecular systems [38, 29, 56, 58] or estimating free energy differences [55, 10].

Such models approximate a target density $\mu$ via diffeomorphic maps $f(\cdot; \boldsymbol{\theta}) \colon \Omega \subset \mathbb{R}^d \to \Omega$ by transforming samples $\boldsymbol{z} \sim p_0(\boldsymbol{z})$ of a base density into samples $\boldsymbol{x} = f(\boldsymbol{z}; \boldsymbol{\theta})$ such that they follow the push-forward density

$$\boldsymbol{x} \sim p_f(\boldsymbol{x}; \boldsymbol{\theta}) := p_0\left(f^{-1}(\boldsymbol{x}; \boldsymbol{\theta})\right)\left|\det \partial_{\boldsymbol{x}} f^{-1}(\boldsymbol{x}; \boldsymbol{\theta})\right|. \tag{1}$$

Flows can be trained on data by maximizing the likelihood or via minimizing of the reverse KL divergence $D_{KL}\left[p_f(\cdot; \boldsymbol{\theta}) \| \mu\right]$ if $\mu$ is known up to a normalizing constant.

While NFs are usually introduced as *smooth* diffeomorphisms, most applications like density estimation or sampling only require $C^1$-smooth transformations. Higher-order smoothness of flows has not been discussed so far and can become a challenge as soon as multi-modal transformations on other topologies than $\mathbb{R}^d$ are discussed.

---

[*]J.K and A.K. contributed equally to this work.

35th Conference on Neural Information Processing Systems (NeurIPS 2021).

$C^k$-smoothness of NFs with $k > 1$ is especially important for applications in physics. Physical models usually come in the form of differential equations, where the derivatives bear a physical meaning that is often crucial to fit, evaluate, or interpret the model. Thus, the construction of expressive, smooth flow architectures will likely open up new avenues of research in this domain.

**Boltzmann Generators**    A recent application of NFs to a physical problem are Boltzmann Generators (BG) [38], which we see as the main application area for the methods introduced in this paper. They are generative models trained to sample conformations of molecules in equilibrium, which follow a Boltzmann-type distribution $\mu(\boldsymbol{x}) \propto \exp(-u(\boldsymbol{x}))$. Here $u$ is the dimensionless potential energy defined by the molecular system and the thermodynamic state (e.g. Canonical ensemble at a certain temperature) it is simulated in. BGs can be trained by a combination of MLE on possibly biased trajectory data and simultaneous minimization of the reverse KL divergence to the target density $\mu(\boldsymbol{x})$. This bi-directional training scheme can achieve sampling of low-energy equilibrium structures. After training, BGs can be used e.g. for importance sampling or for providing efficient proposals when being used in MCMC applications [45].

In the context of BGs, the negative gradient of the log push-forward density with respect to $\boldsymbol{x}$ corresponds to the atomistic forces. Access to well-behaved forces is pivotal in most classical molecular computations: they simplify optimization of molecular models, drive molecular dynamics (MD) simulations, enable computations of macroscopic observables, and facilitate multiscale modeling.

In this work, we will primarily focus on two important implications of smooth flow forces. First, we show that they enable the training of NFs via force matching (FM), i.e. minimizing the force mean-squared error with respect to reference data. In combining FM with density estimation, NFs are pushed to match the distributions and their derivatives, which implicitly leads to favorable regularization. Second, we apply flow forces to drive dynamics simulations. Such simulations are required in many applications to not only sample the target distribution but also compute dynamical observables such as transition rates or diffusivities.

**Respecting Topological Constraints**    Many physical models operate on nontrivial topologies, i.e. the $d$-dimensional hyper-torus $\mathbb{T}^d$, which can become an obstacle in constructing smooth flows.

An important example are BGs for peptides and small proteins which require an internal coordinate (IC) transformation to achieve low-energy sampling of structures [38]. This non-trainable layer transforms Euclidean coordinates $\boldsymbol{x} \in \mathbb{R}^{n \times 3}$ into a representation given by distances $\boldsymbol{d} \in [a_1, b_1] \times \ldots \times [a_{n-1}, b_{n-1}]$, angles $\boldsymbol{\alpha} \in [0, \pi]^{n-2}$ and dihedral torsion angles $\boldsymbol{\tau} \in \mathbb{T}^{n-3}$. As molecular energies are often invariant to translation and rotation, IC transformations are useful as they are bijective with the equivalence class of all-atom coordinates that are identical up to global rotations and translations. The learning problem can then be reduced to modeling the joint distribution $\mu(\boldsymbol{d}, \boldsymbol{\alpha}, \boldsymbol{\tau})$ which is supported on the nontrivial topological space $\mathcal{X}_{\text{IC}} := \mathbb{I}^{2n-3} \times \mathbb{T}^{n-3}$, where $\mathbb{I} = [0, 1]$ denotes the closed unit interval.

Recent work [38, 56] suggested to model the density within an open set $\Omega \subset \mathcal{X}_{\text{IC}}$, leverage $C^\infty$-smooth normalizing flows defined on $\mathbb{R}^d$ and then prevent significant mass around singular points using regularizing losses. Such an approach however can lead to bias and ill-behaved training and requires a-priori knowledge of the support of the densities. Later work [9] approached the problem using $C^1$-smooth splines flows which leads to accurate samples, however results in broken forces.

To overcome these limitations while still benefiting from the merits of prior work, such as bi-directional training and fast forward/inverse transformations we formulate the following desiderata for flow transformations on-top of IC layers: (A) They must have support on $\mathbb{I}^d$ and $\mathbb{T}^d$. (B) They should be $C^\infty$-smooth. (C) They must allow bi-directional training. (D) Forward and inverse direction must be efficient to evaluate.

Satisfying (D) can be achieved using coupling layers [11, 12]. This reduces the problem to finding element-wise conditional transformations satisfying (A-C).

**Contributions**    In this work we propose the following novelties:

- We present a new $C^\infty$-smooth transformation with compact support which can simultaneously be used for expressive transformations on $\mathbb{I}^d$ as well as the hypertorus $\mathbb{T}^d$. This satisfies (A) and (B).

- We present novel algorithm which allows optimizing non-analytic inverse directions of flows that can only be evaluated via black-box root finding methods. This satisfies (C).
- We show that training of smooth flows through combinations of force matching, density estimation, and energy-based training can achieve nearly perfect equilibrium densities.
- We show that forces of such smooth flows can be used in dynamical simulations.

## 2 Related Work

While related work exist for each of the requirements (A-D) above, none of them provides a framework that addresses all of them. Specifically, multi-modality in element-wise flow transformations has been approached using splines [13, 35], monotonoic polynomials [26, 41], monotonic neural networks which directly approximate an inverse CDF transformation [22, 8], mixtures of unimodal transformations [21, 43] and stochastic transition steps [56, 7].

Flows on non-trivial manifolds have been discussed for hypertori and -spheres [43], Lie groups [16, 4], hyperbolic spaces [3] as well as on general manifolds [15, 18, 33, 32, 5, 27]. Applications of flows for molecular systems include approximation of the equilibrium density [38, 56, 29], generation of molecular conformations [58, 44] and free energy differences [55, 10]. Using forces to train neural network potentials of molecules was done e.g. in [53, 24].

Backpropagation through black-box functions was generally discussed in [19]. More related to our work is Bai et al. [2] who discuss training models whose outputs are obtained through fix-point equations. Finally Shirobokov et al. [46] discuss backpropagation through black-box simulators using surrogate models.

Using force-matching for training generative models is rarely used in machine learning due to absence of an (unnormalized) target density. However, a common method to train unnormalized energy-based models purely on data is given by *score-matching* (SM) [25, 47]. SM assumes an unknown density for the data distribution and attempts to match forces implicitly, e.g. via sliced [49] or denoising [52] score-matching. A survey on score-matching and its relation to other approaches of training energy-based models on data is e.g. given in Song and Kingma [48].

We discuss related work on flows for $\mathbb{I}$ and the unit circle in detail in Section 4.

## 3 Incorporating Forces into Flow Training

NFs are most commonly trained by minimizing the negative log likelihood (NLL),
$$\mathcal{L}_{\mathrm{NLL}}(\boldsymbol{\theta}) := -\mathbb{E}_{\boldsymbol{x}\sim\mu(\boldsymbol{x})}[\log p_f(\boldsymbol{x};\boldsymbol{\theta})] = D_{\mathrm{KL}}[\mu||p_f(\cdot;\boldsymbol{\theta})] + \mathrm{const}, \tag{2}$$
or by minimizing the reverse KL divergence (KLD)
$$\mathcal{L}_{\mathrm{KLD}}(\boldsymbol{\theta}) := D_{\mathrm{KL}}[p_f(\cdot;\boldsymbol{\theta})||\mu] + \mathrm{const}. \tag{3}$$
BGs as discussed in Noé et al. [38] use a convex combination of the two in order to avoid mode-collapse while still being able to achieve low-energy samples. If reference forces $\mathbf{f}(\boldsymbol{x}) = -\partial_{\boldsymbol{x}} u(\boldsymbol{x})$ corresponding to samples $\boldsymbol{x}$ are available and $f(\cdot;\boldsymbol{\theta})$ is at least $C^2$-smooth, the optimization can naturally be augmented by the force mean-squared error:
$$\mathcal{L}_{\mathrm{FM}}(\boldsymbol{\theta}) := \mathbb{E}_{\boldsymbol{x}\sim\mu(\boldsymbol{x})}\left[\|\mathbf{f}(\boldsymbol{x}) - \partial_{\boldsymbol{x}} \log p_f(\boldsymbol{x};\boldsymbol{\theta})\|_2^2\right]. \tag{4}$$
As was shown in Wang et al. [53] such *force-matching* can lead to unbiased potential surfaces even if samples are not sampled from equilibrium. Similarly to Noé et al. [38] we can thus define a loss function for smooth flows as the convex combination
$$\mathcal{L}(\boldsymbol{\theta}) = \omega_n \mathcal{L}_{\mathrm{NLL}}(\boldsymbol{\theta}) + \omega_k \mathcal{L}_{\mathrm{KLD}}(\boldsymbol{\theta}) + \omega_{\mathrm{f}} \mathcal{L}_{\mathrm{FM}}(\boldsymbol{\theta}). \tag{5}$$

## 4 Smooth flows on the closed interval and the unit circle

We now discuss how to achieve smooth flows on $\mathbb{I}^d$ and $\mathbb{T}^d$. To unify the discussion we consider the unit circle $S^1$ to be the quotient space $\mathbb{I}/\sim$ using the relation $x \sim x' \Leftrightarrow (x = x') \vee (x = 0 \wedge x' = 1) \vee (x' = 0 \wedge x = 1)$. The $d$-dimensional hypertorus is given by the direct product $\mathbb{T}^d = S^1 \times \ldots \times S^1$. Following the usual definition we say that $f$ is $C^k$-smooth on a compact interval $[a, b]$ iff there exists a $C^k$-smooth continuation of $f$ on an open set $\Omega \supset [a, b]$.

**Smooth flows on $\mathbb{I}$** Any smooth diffeomorphism on $\mathbb{R}$ or an open interval $(a, b) \supset \mathbb{I}$ can be restricted to $\mathbb{I}$ and re-scaled to satisfy this requirement. Let $f: \Omega \to \Omega$ be $C^k$-smooth for some open set $\mathbb{I} \subset \Omega \subset \mathbb{R}$. Then $\tilde{f}(x) = (f(x) - f(0))/(f(1) - f(0))$ defines a $C^k$-smooth diffeomorphism on $\mathbb{I}$. Simple $C^\infty$ transformations on $\mathbb{R}$ are given by affine transformations [12]. After re-scaling any affine map will result in the same identity mapping and thus will not be able to model complex densities. Powerful and computationally efficient multi-modal transformations on $\mathbb{I}$ can be achieved using rational-quadratic splines [13]. Those are $C^1$-smooth and thus will result in discontinuous forces which can be disadvantageous for physical applications. Possible other $C^\infty$-smooth candidate transformations with analytic forward evaluation are given by mixtures of logistic transformations [21] or deep sigmoid/deep dense sigmoid flows [22]. Those are only continuous on $(0, 1)$ and thus special care has to be taken to avoid problematic behavior on the tails. Finally, multi-modal $C^\infty$ transformations on $\mathbb{R}$ can be achieved using non-analytic methods [6, 54]. Yet, computational costs (solving and backpropagating over ODEs, inverting quadrature integrations via bisections) and numerical accuracy (non-anlytic forward passes) limit their applicability e.g. when used in deep coupling layers which are required to capture multivariate correlations or when trying to match densities up to high accuracy e.g. as necessary for molecular modeling.

**Smooth flows on $S^1$** Flows on the hypertorus were discussed in Rezende et al. [43] who introduced necessary conditions for the transformations to define $C^0$-continuous densities, such that they can be used in density estimation tasks. In their work Rezende et al. [43] introduced three candidates satisfying this condition: mixtures of non-compact projections (NCP), rational-quadratic circular spline flows (CSF) and mixtures of Moebius transformations (MoMT). While NCPs and CSFs satisfy $C^1$-continuity and thus render forces discontinuous, only MoMTs define smooth densities. While MoMTs trivially also define $C^\infty$-flows on $\mathbb{I}$ their periodicity would be limiting when applied to non-periodic densities.

**Smooth compact bump functions** In addition to this previous work we leverage a third way to construct smooth transformations which works for both $\mathbb{I}$ and $S^1$. Here we follow a general construction principle for smooth bump functions e.g. as explained in Tu [51]. All proofs can be found in the supplementary material.

First, define a $C^k$-smooth and strictly increasing ramp function $\rho: \mathbb{I} \to \mathbb{I}$ with $\rho(0) = 0$ and $\rho(1) = 1$. Second, define the generalized sigmoid

$$\sigma[\rho](x) := \frac{\rho(x)}{\rho(x) + \rho(1 - x)}. \tag{6}$$

Then $\sigma[\rho]$ will be a $C^k$-diffeomorphsim on $\mathbb{I}$ and furthermore all derivatives up to order $k$ vanish at the boundary. The first derivative defines a non-negative smooth bump function with support $[0, 1]$ and maximum at 0.5. We can introduce a concentration parameter $a \in \mathbb{R}_{>0}$ and a location parameter $b \in [0, 1]$, which makes $g(x) := \sigma[\rho]\left(a \cdot (x - b) + \frac{1}{2}\right)$ a smooth bijection from $[-\frac{1}{2a} + b, \frac{1}{2a} + b]$ to $[0, 1]$ with all values and higher-order derivatives vanishing outside the domain. Furthermore, by introducing $c \in (0, 1]$ and setting

$$f(x) := (1 - c) \cdot \left(\frac{g(x) - g(0)}{g(1) - g(0)}\right) + c \cdot x, \tag{7}$$

we can define flexible unimodal $C^k$-diffeomorphisms on $[0, 1]$. When used within coupling layers, we let $a, b, c$ be the output of some not further constrained neural network.

**$C^k$ and $C^\infty$ ramp functions** There are many possible choices for ramp functions satisfying above mentioned properties. A simple $C^k$ ramp function is given by the $k$-th order monomial $\rho(x) = x^k$. More interestingly, $C^\infty$ smoothness on $[0, 1]$ can be achieved using the ramp $\rho(x) = \exp\left(-\frac{1}{\alpha \cdot x^\beta}\right)$ for $x > 0$ and $\rho(x) = 0$ for $x \leq 0$. Here we set $\alpha > 0$ and $\beta > 1$. We make $\alpha$ a trainable parameter. Optimizing $\beta$ is possible in principle, yet fixing $\beta \in \{1, 2\}$ and treating it as a hyper-parameter stabilized training and led to better results.

**Smooth and efficient circular wrapping** As discussed in Rezende et al. [43] and Falorsi et al. [16] we can turn any $C^k$-smooth density $p(x)$ with support on $\mathbb{R}$ into a $C^k$-smooth density on $S^1$ using the marginalization $\tilde{p}(x) = \sum_{k \in \mathbb{Z}} p(x + k)$. This construction is generally problematic due to two

reasons: (i) non-vanishing tails (e.g. if $p(x)$ is Gaussian) require evaluating an infinite series which in most cases has no analytic expression and can be numerically challenging (ii) smooth densities with compact support in some interval $[a, b]$, for example sigmoidal transforms, can introduce non-smooth behavior at the interval boundaries.

Smooth and compactly supported transformations as introduced above do not suffer from this: (i) due to compact support the series will always have a finite number of non-vanishing contributions, (ii) as the transformations are $C^k$ on all of $\mathbb{R}$ and all derivatives are compactly supported wrapping will always result in a $C^k$-smooth density on $S^1$.

**Multi-modality via mixtures** Similarly, as discussed in prior work [21, 43] any convex sum of $C^k$-diffeomorphisms on $\mathbb{I}$ defines a $C^k$-diffeomorphism on $\mathbb{I}$. Thus we can combine multiple of those unimodal transformations defined in Eq. (7) to obtain arbitrarily complex multi-modal transformations on $\mathbb{I}$ and $S^1$, see Fig. 1.

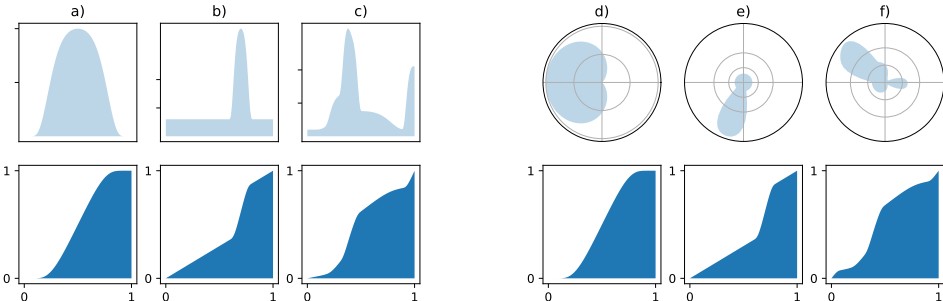

Figure 1: Construction of mixture transformations on compact intervals (left) and hypertori (right). The upper and lower row depict probability densities and cumulative distribution functions, respectively. Multiple unimodal bump functions [a) and d)] are added to a small but finite density [b) and e)] and combined to yield bijective multimodal transformations [c) and f)].

# 5 Optimizing non-analytic inverse flows

A drawback of mixture-based flows is their lack of an analytic inverse. Prior work such as [21, 43] suggested to rely on black-box inversion methods like a bisection search to sample from trained models. While this leads to accurate samples, the discrete nature of such black-box oracles does not allow to minimize $\mathcal{L}(\boldsymbol{\theta})$ as defined in Eq. (5).

A possible remedy to this can be derived using the *inverse function theorem*. Let $f(\cdot; \boldsymbol{\theta}): \Omega \subset \mathbb{R} \to \Omega$ be a scalar diffeomorphism and let furthermore $x = f^{-1}(y; \boldsymbol{\theta})$ be obtained via a black-box inversion algorithm. To minimize losses depending on $x(y; \boldsymbol{\theta})$ or $\log |\partial_y x(y; \boldsymbol{\theta})|$ we need to compute gradients with respect to $y$ and $\boldsymbol{\theta}$. Here we derive the following relations for the derivatives (all details in supplementary material):

$$\partial_y x(y; \boldsymbol{\theta}) = (\partial_x f(x; \boldsymbol{\theta}))^{-1} \tag{8}$$

$$\partial_{\boldsymbol{\theta}} x(y; \boldsymbol{\theta}) = - (\partial_x f(x; \boldsymbol{\theta}))^{-1} \partial_{\boldsymbol{\theta}} f(x; \boldsymbol{\theta}) \tag{9}$$

$$\partial_y \log |\partial_y x(y; \boldsymbol{\theta})| = - (\partial_x f(x; \boldsymbol{\theta}))^{-1} \log |\partial_x f(x; \boldsymbol{\theta})| \tag{10}$$

$$\partial_{\boldsymbol{\theta}} \log |\partial_y x(y; \boldsymbol{\theta})| = - (\partial_x f(x; \boldsymbol{\theta}))^{-1} (\log |\partial_x f(x; \boldsymbol{\theta})| \partial_{\boldsymbol{\theta}} f(x; \boldsymbol{\theta}) - \partial_{\boldsymbol{\theta}} \partial_x f(x; \boldsymbol{\theta})) \tag{11}$$

We remark that (8) corresponds to the *implicit reparameterization gradient* as introduced in Figurnov et al. [17]. Using these rules, we can first compute $x$ via a black-box oracle for any input $y$ and then compute all necessary gradients using forward evaluations and derivatives of $f(\cdot; \boldsymbol{\theta})$. Derivatives of $f(\cdot; \boldsymbol{\theta})$ are usually accessible for computing the log Jacobian of the transformation. We can obtain higher-order derivatives using automatic differentiation. It is easy to see that this construction extends to multivariate diffeomorphisms with diagonal Jacobian as used in coupling layers. It can be generalized to arbitrary higher order derivatives using the Faà di Bruno formula, e.g. when aiming

to do force matching through a black-box inversion algorithm. As our experiments only require losses involving second order derivatives for the inverse direction we leave this for future work. Note that Equations (8) - (11) can be applied to any other element-wise flow transformation that lacks an analytic inverse and is not tied to mixture models.

Another problem of bisection is its sequential execution which can become prohibitively slow during training. E.g. achieving an error of $10^{-6}$ requires around 20 iterations. To obtain speedup on GPUs we suggest to generalize the classic binary search to searching in a grid of $K$ bins simultaneoulsy (see details in Supp. Mat.). For $K = 2$ it results in the usual bisection method. However, by leveraging vectorized execution we can reduce the number of iterations by a factor $O\left(1/\log K\right)$ at the expense of increasing memory by a factor $O\left(K\right)$. Practical speedup depends on the actual number of parallel workers and thus depending on dimensionality, batch size and number of mixture components the optimal choice of $K$ varies.

## 6 Experiments

The numerical experiments in this work are tailored to show the benefits of smooth flows over non-smooth flows. Therefore, we mainly compare mixtures of bump functions to neural spline flows [13], which exhibit state-of-the-art generative performance but whose densities are only first-order continuously differentiable.

### 6.1 Illustrative Toy Example

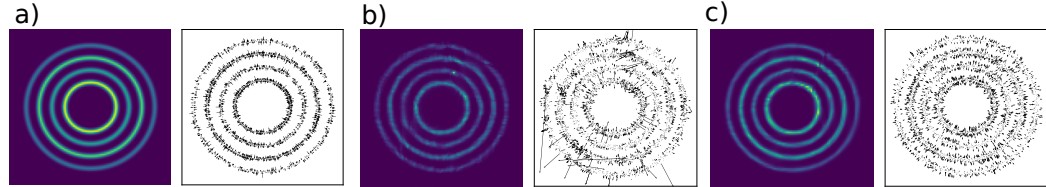

Figure 2: a) Reference density and corresponding force field as approximated by b) a $C^1$-smooth NSF and c) a $C^\infty$-smooth flow using the mixture of bump functions introduced in Sec. 4.

To highlight the difference in terms of smoothness, the two flow architectures were applied to a two-dimensional toy example. Fig. 2 a) depicts the reference energy and forces on samples from the ground-truth distribution. Both flows were trained through density estimations on those samples (see SI for details). The smooth flows and spline flows matched the density well, which demonstrates their expressivity. However, the force field of the spline flow contained dramatic outliers, while the forces of the smooth flow matched the regular behavior of the reference forces.

### 6.2 Runtime Comparison

While the rational-quadratic splines are analytically invertible, mixture transformations obviously increase computational cost of the inversion due to the iterative root-finding procedure. To quantify this gap in performance, the inverse evaluation of a spline flow was compared with a modified version, where the analytic inverse was replaced by the multi-bin bisection from Section 5. The performance of the bisection was evaluated for different numbers of bins $K = 2^m$, $m = 1, \ldots, 8$, and the most efficient $K$ was picked for each input size ("dim"). Both transformations were coupled to a conditioner with the same input size (dim) and 64 hidden neurons.

For small tensor dimensions (2-32), the optimal multi-bin bisection employed up to 256 bins on the GPU, which resulted in only a factor of 2-3 slowdown compared to analytic inversion. For larger dimensions (2048), the parallelization over multiple bins became less effective, leading to one order of magnitude difference in computational cost. The compute times and optimal bin sizes were comparable for the inverse network pass and its backward-pass. More details on these runtime comparisons can be found in the supplementary material.

## 6.3 Smooth Flows Enable Boltzmann Generator Training through Force Matching

To demonstrate the advantages of smooth flows, we train a Boltzmann generator (BG) for a small molecule, alanine dipeptide, which is described in the supplementary material. This is a common test case for molecular computations and was previously considered for sampling with normalizing flows in [56, 9, 30]. The molecular system has 60 degrees of freedom (global rotation and translation excluded). Its potential energy surface is highly sensitive to small displacements [30] and contains singularities.

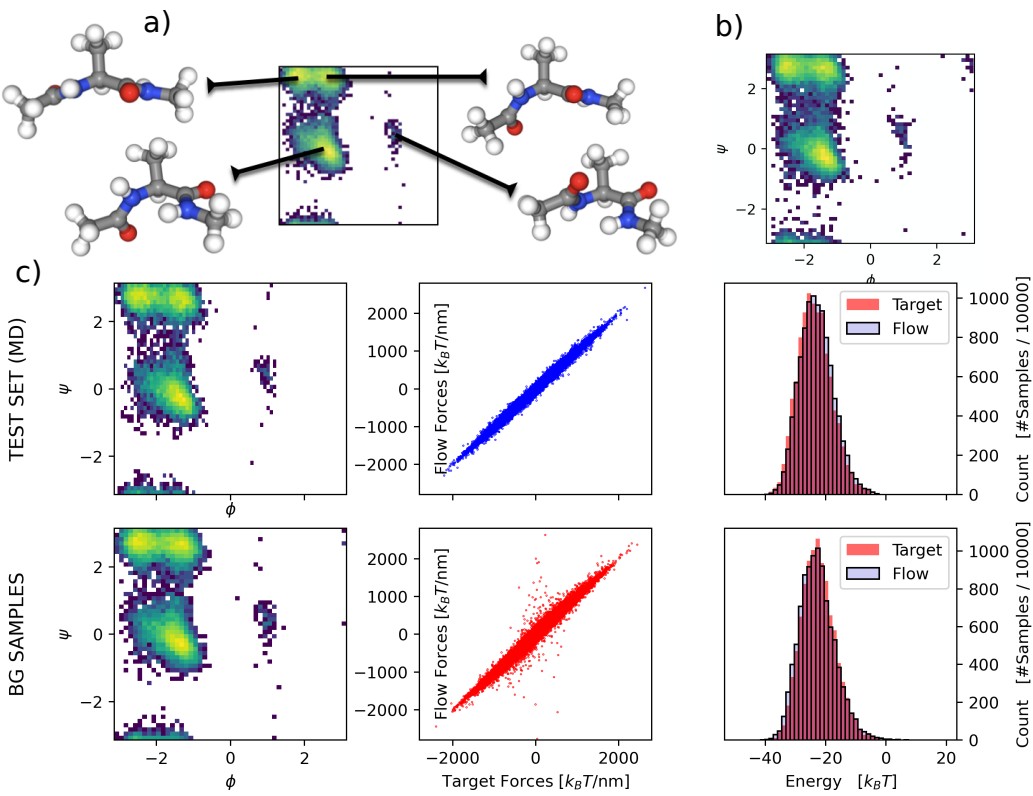

Figure 3: a) Generated sample structures. b) Torsion distribution after training through root-finding. c) Smooth normalizing flow trained on alanine dipeptide through a combination of force matching and density estimation. The top and bottom row show the performance on 10,000 samples each from the holdout test set and from the flow, respectively. Left: joint distribution of backbone torsion angles. Center: scatter plot of flow vs. target force components. Right: energy histograms for the flow energy and the target energy. (Flow energies were shifted by a constant so that the minimum energy matched with the minimum energy from the molecular potential).

In previous work [56, 9, 30], BGs for alanine dipeptide used stochastic augmentation of the base space. The introduction of noise variables facilitates creating expressive normalizing flows but prevents computation of deterministic forces. Some have also used spline flows to sample molecular configurations [55, 10].

As a proof-of-concept for the force matching loss, we trained smooth flows for alanine dipeptide using a 1000:1 weighting between the force matching residual and the negative log likelihood, see supplementary material for details on the flow and training setup. No stochastic augmentation or energy-based training was used and no reweighting was conducted for postprocessing.

Figure 3 c) compares the flow distribution with the target Boltzmann distribution on the test data from MD and on flow samples. The left column shows that the flow has almost perfectly learned the nontrivial joint distribution between the two backbone torsion angles $\phi$ and $\psi$, including the sparsely populated metastable region around $\phi \approx 1$. In contrast to previous work that used affine coupling layers [56], the modes are cleanly separated, see also Fig. SI-3 for a direct comparison. The center column compares flows forces with target forces. Those forces are highly sensitive to small

perturbations of the molecular configurations. Nevertheless, they matched up to a root-mean square deviation of 25 and 46 $k_BT$/nm in the target ensemble and the generated ensemble, respectively. Note that the training was stopped after 10 epochs, where the validation force matching residual had not yet fully converged, so that even further improvements may be possible with longer training or extensive hyperparameter optimization. A more detailed analysis of the forces and sampling efficiency of the BGs is shown in supplementary material, Figure SI-4. Smooth flows trained by a combination of density estimation and FM attain a favorable sampling efficiency of 42%, compared to 25% and < 1% for spline and RealNVP flows, respectively.

Finally, the right column of Figure 3 c) depicts the distribution of flow and target energies evaluated on the same samples. The flow energies were shifted so that the minimum energy matched with the molecular mechanics energy. It has not escaped our notice that this constant offset (189 $k_BT$ on the test set and 190 $k_BT$ on the flow samples) corresponds to the log partition function, a quantity whose intractability has complicated research in statistical mechanics for decades.

The energy distribution of the flow tracked the ground truth exponential distribution almost perfectly even though no target energies or forces were evaluated on the flow samples during training. This close-to-perfect match demonstrates that including force information into the training process presents an efficient means for regularization. Figure 3 a) shows representative conformations generated by the BG, which confirm the high quality of the molecular structures.

### 6.4 Boltzmann Generator Training through Black-Box Root-Finding

Table 1: Test set metrics of alanine dipeptide training with different losses: negative log likelihood (NLL), force matching error (FME), and reverse Kullback-Leibler divergence (KLD). Weighted loss functions were used as defined in Eq. (5) with weight factors $\omega_k$, $\omega_f$ and $\omega_n = 1 - \omega_k - \omega_f$. Metrics were recorded after 10 training epochs. Statistics are means and $2\times$ standard errors over 10 replicas for each experiment. The lowest value with respect to each metric is highlighted in bold type.

| Metric | Method | $\omega_k = 0$ $\omega_f = 0$ | $\omega_k = 0$ $\omega_f = 0.001$ | $\omega_k = 0.1$ $\omega_f = 0$ | $\omega_k = 0.1$ $\omega_f = 0.001$ |
|---|---|---|---|---|---|
| NLL | spline | -210.32 ($\pm$0.16) | - | -196.40 ($\pm$ 1.53) | 48.13 ($\pm$ 95.37) |
| | smooth | -211.04 ($\pm$ 0.09) | **-211.40** ($\pm$ 0.05) | -206.33 ($\pm$2.04) | -208.70 ($\pm$ 1.93) |
| FME $\times 10^4$ | spline | 12.13 ($\pm$ 4.94) | - | 313.64 ($\pm$ 107.92) | 1498.98 ($\pm$ 1935.72) |
| | smooth | 1.04 ($\pm$ 0.08) | **0.32** ($\pm$ 0.02) | 5.80 ($\pm$ 2.30) | 0.48 ($\pm$ 0.05) |
| KLD | spline | 263.77 ($\pm$ 3.21) | - | 230.08 ($\pm$ 12.38) | 1205.69 ($\pm$ 36.25) |
| | smooth | **193.91** ($\pm$ 0.39) | 195.17 ($\pm$ 0.43) | 219.03 ($\pm$ 22.16) | 207.81 ($\pm$ 11.64) |

Two experiments were conducted to demonstrate that flows can be trained through a black-box root-finding algorithm.

First, the BG from section 6.3 was trained with all smooth transformations operating in the inverse direction. Consequently, inverse problems had to be solved when computing the negative log likelihood with respect to data, while the sampling occurred analytically. Figure 3 b) shows the joint marginal distribution of the backbone torsions on the BG samples. The distribution matches the data and the smooth flows that were trained in forward mode (see Fig. 3). This result shows that training with gradients computed from the inverse function theorem is feasible.

Second, flows were trained by different combinations of density estimation, force matching, and energy-based training, where computing $\partial_\theta \mathcal{L}_{\text{KLD}}$ requires the gradients of the inverse flow. The training of spline flows with inclusion of the force matching error ($\omega_f > 0$) was notoriously unstable. This led to 10/10 and 7/10 failed runs for $\omega_k = 0$ and $\omega_k = 0.1$, respectively. This is understandable

from the discontinuity of the spline forces. Even the combination of NLL and KLD led to instabilities for 2/10 runs. (Gradients were not clipped during training to enable a mostly unbiased comparison). In contrast, all training runs with our smooth flows concluded successfully.

Table 1 shows the metrics on the test set after 10 training epochs using spline flows and smooth flows (see SI for specifics about the flow architecture and training). With pure density estimation ($\omega_k = \omega_f = 0$), both architectures achieved similar NLL. However, the smooth flow achieved much lower FME and KLD. The NLL and FME were further improved when force data was presented to the flow during training. Including reverse KLD in the loss yielded reasonable metrics for the smooth flows, indicating that the backpropagation through root-finding was numerically stable. However, the metrics were consistently worse than with pure NLL training for both types of flows (spline and smooth) and were subject to large fluctuations. In contrast, including the FME proved to be a more stable approach to include information about the target energy into the training.

## 6.5 Smooth Flows as Potentials in Molecular Dynamics

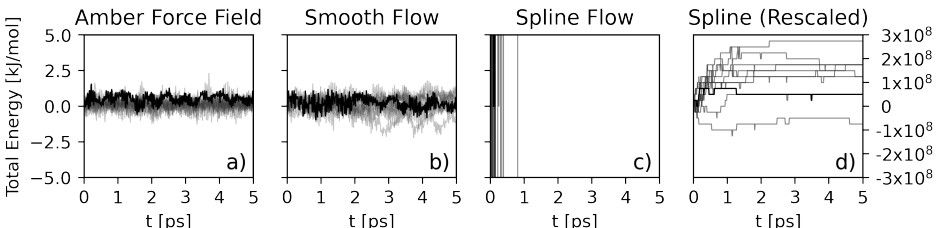

Figure 4: Total energy fluctuation in simulations with a classical force field and with two flows that employed smooth transforms and neural spline transforms, respectively. For each potential, 10 simulations were run starting from 10 different initial configurations (grey color). One run each is highlighted in black. Subfigures c) and d) show identical data on different scales.

A challenging test for the smoothness of a flow is its use as a potential energy in a dynamics simulation. Especially simulations in the microcanonical ensemble are extremely sensitive to numerical errors as the symplectic integration does not impose any additional stabilizing mechanism on the total energy. This means that any inconsistencies between energies and forces on the scale of one integration step easily cause drifting, strongly fluctuating, or exploding energies.

Therefore, we ran MD simulations using the flow energy $u_\theta = -\log p_f(\cdot; \boldsymbol{\theta})$, from Section 6.3 as the potential. For comparison, simulations were also run with spline flows that were trained purely by density estimation. Simulations started from stable structures of an MD simulation and were equilibrated in each potential for 1 ps using a Langevin thermostat with 10/ps friction coefficient.

Figure 4 depicts the evolution of the total energy over 5 ps simulations in the microcanonical ensemble. As expected, the classical simulation kept the total energy roughly constant within a standard deviation of $6 \times 10^{-4}$ kJ/mol per degree of freedom. Astoundingly, the smooth flow potentials also maintained the energies within $8 \times 10^{-4}$ kJ/mol per degree of freedom using the same 1 fs time step. In contrast, the simulations with spline flow potentials quickly fell apart with potential and kinetic energies growing out of bounds. Those instabilities persisted even with an order-of-magnitude smaller time step (not shown).

While the infeasibility of spline flows for this task was expected, the competitive behavior of smooth flows with molecular mechanics force fields that were tailored for dynamics highlights their regularity and the consistency of their forces.

## 7 Discussion

**Limitations and outlook** Despite the promising results we mention the following possible limitations of the method in the present state and discuss possible improvements for future work:

First, smooth flows as implemented in this work impose numerical overhead in comparison to non-smooth alternatives such as spline flows. While this can be considered a question of concrete engineering it also opens search for more efficient smooth alternatives.

In addition, the differentiation rules for black-box inverses provide correct algebraic gradients. Yet for considerably multi-modal distributions maintaining numerical stability can become a challenge. Future work should thus focus on studying and improving numerical stability of the bidirectional training.

While potentials can be trained up to an accuracy where they allow for energy-conserving simulations, it is yet to show that they also provide accurate equilibrium distribution for long-run simulations. At this point a major bottleneck is the inferior evaluation performance per step of a flow-based force-field compared to common force-field implementations.

Furthermore, internal coordinates are a very efficient representation space for smaller poly-peptides. However, they are difficult to scale to large proteins, protein complexes or systems with non-trivial topologies, e.g. proteins with disulfide bonds. Integrating recent advances in graph-based molecular representations with force-matched flows will likely be required for a succesful modeling of such systems.

Finally, training with forces improves the resulting potential compared to just training with samples. However, for many MD data forces have not been stored and recomputing them requires a significant computational overhead.

**Conclusion**    We introduced a range of contributions which can improve upcoming work on applying flows to physical problems.

The proposed approach for backpropagation through black-box root-finders can help to obviate the search for analytically invertible transformations if bi-directional training is necessary. As we show the method works especially well for relatively low-dimensional problems. Scaling the approach, generalizing it to non-diagonal Jacobians and improving its numerics are interesting questions for future work.

The MD simulation example shows that $C^\infty$-smooth flows on non-trivial topologies can open new avenues of research as well as new applications for flows. By carefully respecting the topological domain as well as the smoothness of the target potential we can approximate the equilibrium density of a small peptide nearly perfectly. Finally, we showed that incorporating force information together with MLE outperforms the state-of-the-art approach of combining MLE with minimizing of the reverse KL divergence. It does not suffer from mode-collapse, yet includes information of the target potential which is required to achieve low-energy samples.

Combined, these results could improve methods used for learning the mean potential surface of coarse-grained molecules, pave the way to multi-scale flows for modeling large protein systems and eventually help to accelerate simulations and sampling.

## Acknowledgements

We thank the anonymous reviewers at NeurIPS 2021 for their insightful comments. Special thanks to Yaoyi Chen for setting up the alanine dipeptide system and Manuel Dibak, Leon Klein, Oana-Iuliana Popescu, Leon Sixt, and Michael Figurnov for helpful discussions and feedback on the manuscript and pointers to related work. We acknowledge funding from the European Commission (ERC CoG 772230 ScaleCell), Deutsche Forschungsgemeinschaft (GRK DAEDALUS, SFB1114/A04), the German Ministry for Education and Research (Berlin Institute for the Foundations of Learning and Data BIFOLD), and the Berlin Mathematics center MATH+ (Project AA1-6 and EF1-2).

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
