# OpenReview forum: "Smooth Normalizing Flows"
_NeurIPS.cc/2021/Conference — NeurIPS 2021 Poster_

### Official Review · Reviewer_TKYQ · 2021-07-12

**Rating:** 6
**Confidence:** 5

**Summary:**

The authors introduce a flow architecture, which they call smooth normalizing flows, that can operate on compact intervals and hypertori and has a C-infinity-smooth transformation. Furthermore, they propose to train normalizing flows via force matching, i.e. matching the gradients of the flow transformation which those of the energy function of the target distribution. They also show that a transformation with non-analytic inverse can be used to define a flow model which is trained by evaluating the inverse via black-box root finding and computing the gradients through the forward pass of each layer.
On the experiment side, the authors apply their method to a toy example as well as the Boltzmann distribution of a complex molecule. They demonstrate that their flow architecture is expressive and the presented training procedure can be successfully used. When force matching is used, the gradients of the flow transformation resemble the respective gradients of the energy function of the target indeed more accurately and they can even be used as forces in molecular dynamics simulations.

**Limitations And Societal Impact:**

Although the authors are focused on approximating Boltzmann distributions, their framework is in theory not limited to this setting so it would be interesting to see how well other densities could be approximated.

**Main Review:**

The authors are motivated by tackling the challenging problem of fitting the Boltzmann distribution of complex molecules. With their work, they made two significant contributions, which overcame problems previous articles had:
(1) They proposed a flow architecture based on mixtures of smooth bump functions, which are C-infinity-smooth, act on a compact interval, and can easily model multimodal distributions. Furthermore, they showed that these kinds of flows can be trained although their inverse is non-analytic by evaluating the inverse through black-box root finding and computing the gradients via the push-forward density.
(2) They trained their models via force matching which makes the gradients of the transformation closer to those of the energy function of the target. Furthermore, this stabilizes training when using the reverse Kullback-Leiber divergence as objective an allows the model to be used as potential in a molecular dynamics simulation.
The first contribution is novel and although force matching is a known training method, it has not been used in this context yet so the second contribution is novel as well.
The experimental results are impressive. However, there should be more comparisons to competing methods. For instance in section 6.3, there is no clear comparison to training without force matching. They mention a competing method, i.e. stochastic normalizing flows, but they only refer to the article and do not show the results when stochastic normalizing flows are applied to the same problem.
Overall, the paper is well written and has clear illustrations.

**Time Spent Reviewing:**

3

---

> ### Author Response · Authors · 2021-08-06
> **Answer to Reviewer TKYQ**
>
> Dear reviewer,
>
> thank you a lot for your time. Please find our answers to your remarks below:
>
> > However, there should be more comparisons to competing methods. For instance in section 6.3, there is no clear comparison to training without force matching.
>
> This is a very good idea. In fact, a comparison of different training approaches is presented in Sec. 6.4 but the results are not visualized. We agree that a visualization of the forces and energies obtained from training without force matching would be helpful and we will happily incorporate these plots in the final version.
>
> > They mention a competing method, i.e. stochastic normalizing flows, but they only refer to the article and do not show the results when stochastic normalizing flows are applied to the same problem.
>
> Reviewer QQCj asked a very similar question. We will copy our answer here:
>
> “SNFs won’t work with force matching and don’t provide tractable model energy. Running several MD steps between each flow layer also adds tremendously to the computational complexity of the method. Furthermore, they provide an orthogonal contribution that can easily be combined with smooth flows. In the SNF paper, they are used together with affine coupling layers and NSF couplings. So combining them with our smooth flow couplings would be straightforward, yet not providing new insight in this context..”
>
> As suggested by Reviewer q1W2 we will add some comparing figures for Sec. 5.
>
> > Although the authors are focused on approximating Boltzmann distributions, their framework is in theory not limited to this setting so it would be interesting to see how well other densities could be approximated.
>
> Smooth flows introduce a specific inductive bias for domains where smoothness is relevant. Beyond the physical examples, as discussed, we are currently not aware of very different problems where higher-order of smoothness is critical for success. In classic density estimation problems, this inductive bias might not be necessary or even harmful (e.g. if the data distribution is non-smooth).

---

> > ### Comment · Reviewer_TKYQ · 2021-08-25
> > **Reply to Paper4357 Authors**
> >
> > Thank you for addressing my concerns. I still believe there might be other interesting problems, such as knowledge distillation, which could be tackled using this approach. However, I acknowledge that the main scope of this article is applying it to Boltzmann distributions so I still think this paper is worth an acceptance.

---

### Official Review · Reviewer_q1W2 · 2021-07-15

**Rating:** 6
**Confidence:** 2

**Summary:**

The paper introduces a class of $C^k$-smooth ($k>1$) transformations working on compact intervals and hypertori that can be used to construct normalising flows.
Such type of smooth transformations are important in physical systems as (1) they respect topological constraints and (2) higher-order derivatives have physical interpretations; however, existing transformations either don't respect the constraints or are limited to have only $C^1$-smoothness.
Learning flows consisting mixtures of proposed transformations that requires black-box root-finders is made efficient through gradients derived by the inverse function theorem.
The benefit of accessing to higher-order derivatives is demonstrated by training normalizing flows with force-matching regularisation and dynamics simulations of trained flows.

**Limitations And Societal Impact:**

There is no single paragraph or section dedicated to limitations.
I suggest the author(s) to add one, along with future works.

There is no societal impact either but I don't think it's applied to this paper as it is a technical manuscript.

**Main Review:**

Originality:
The proposed transformations are novel and the application of the inverse function theorem is neat.
This work clearly fills the gap of $C^k$-smooth ($k>1$) flows on intervals and hypertori and provides necessary discussion on related works in the end of section 1.

Quality:
The paper is a complete piece and the techniques proposed (both transformations and training method) are technically sound.
The intuition behind each transformation is given when it is introduced,
and the experiments clearly shows that the proposed method can be trained successfully while having well defined higher-order derivatives.
The authors mostly focus on the advantages of smooth flows in compare to spline flows but limitations are not adequately discussed (only discussion on computation cost is given).

Clarity:
The paper is well-written: the motivation, how the transformations and training are explained and how the method is evaluated are all clear and the structure makes sense.
While it might be due to page limit, it could be helpful to visualise each transformations (and their alternatives) when they are introduced and contrasted in section 4.
In section 5, it was mentioned in L245 that "In contrast to previous work that used affine coupling layers [43], the modes are cleanly separated." but not visualisation was shown.
Again it would be helpful to see it in the paper.

Significance:
The results clearly show on $C^k$-smoothness can help model training (section 6.4) and model usage (section 6.5).
As said, it is not achievable by previous methods.
As motivated and demonstrated by the authors, smooth flows can be widely used in modelling physics systems in which topological constraints are needed, so I can see smooth flows would be beneficial to the community especially who works on scientific modelling.

Misc:
- How would other types of generative models perform on the toy example in section 6.1, or in other experiments? For example I wonder if [1] would a strong competitor as it can learn the support and the distribution on it quite well. Extra experiments would be great to have but no necessary I believe, but it's worth some discussion on why flows are the right choice of generative models in the beginning.

[1] Arbel, Michael, Liang Zhou, and Arthur Gretton. "Generalized energy based models." arXiv preprint arXiv:2003.05033 (2020).

**Time Spent Reviewing:**

3

---

> ### Author Response · Authors · 2021-08-06
> **Answer to Reviewer q1W2**
>
> Dear reviewer,
>
> thanks a lot for your time. Please see our answers to your remarks below:
>
> >  but limitations are not adequately discussed (only discussion on computation cost is given).
>
> We will add a specific section in the final manuscript where we will discuss the limitations in detail. These will include computational costs, as well as discussing the specific inductive bias of the method and challenges that are still a big problem for future work. We will further explain more clearly strengths and weaknesses compared to other candidate flow architectures.
>
> > While it might be due to page limit, it could be helpful to visualise each transformations (and their alternatives) when they are introduced and contrasted in section 4
>
> This sounds like a very good suggestion! We will add such an explanatory visualization to the final manuscript.
>
> > In section 5, it was mentioned in L245 that "In contrast to previous work that used affine coupling layers [43], the modes are cleanly separated." but not visualisation was shown. Again it would be helpful to see it in the paper.
>
> This is a valid point as well! We will add a comparative plot.
>
> > How would other types of generative models perform on the toy example in section 6.1, or in other experiments? [...] Extra experiments would be great to have but no necessary I believe, but it's worth some discussion on why flows are the right choice of generative models in the beginning.
>
> While pure density estimation would be possible with EBMs and similar models (e.g. like the model referred to by you) our task slightly differs as we already know the target energy (up to its normalizing constant). Thus an EBM-like model will likely not give us more information than we already have. In addition, sampling from such an EBM-like model (if trained to proper accuracy) would be as hard as sampling from the original Boltzmann distribution.
>
> Other generative models, such as latent variable models (e.g. VAEs) could be used as well for density estimation together with variational inference as we do in this work. The referenced stochastic normalizing flow [43] would be such a model. However, due to the intractability of the model energy (and forces) certain applications, such as molecular dynamics simulations of the learned model, would not be possible. As explained in answers to the other reviewers, it would easily be possible to combine the proposed smooth flows with such latent variable models, yet we do not think that adding such a comparison would contribute meaningfully to the questions of this work.
>
> We will try our best to explain this more clearly in the final manuscript and motivate why flows are filling a sweet spot here.

---

> > ### Comment · Reviewer_q1W2 · 2021-08-16
> > **Thanks for the author response**
> >
> > Thanks for the response.
> > I will keep my score as it's not possible for me preview any update from the author(s), but assuming the author(s) will update the clarification points and discussions as promised, the paper is worth an acceptance.

---

### Official Review · Reviewer_9eZb · 2021-07-16

**Rating:** 7
**Confidence:** 3

**Summary:**

The paper considers a set of problems that come up when fitting normalizing flows to datasets of interest in physical sciences, and proposes a set of solutions to those problems.

To begin with, when it comes to datasets in physical sciences we might be interested in matching the *gradient* of the log density of the target distribution, as well as the log density itself. This is because this gradient can correspond to the *forces* acting on the physical objects, and as such the gradient is important to get right, in order to make the model useful in downstream applications, e.g. simulation. Furthermore, we often have access to the *true* forces at the points in the training sample, and would like to regularize the model to make its gradients match the available true gradients. Authors consider a straightforward way to implement such regularization: adding the MSE between the model's gradient and the true gradient to the training objective. However, doing this requires a normalizing flow model that is at least $C^2$-smooth, as the second-order gradients will be used during optimization.

Moreover, the data of interest in this context often has non-trivial topology, containing quantities such as angles, directions or axis. There's been prior work on building normalizing flows suitable for such data, but authors point out that there is no existing model that would be
a) restricted to a manifold of interest (say, a circle or a torus)
b) $C^{\infty}$ (or at least $C^k$ for some $k > 1$) continuous
c) *expressive*, i.e. able to represent complex densities on the manifold

Authors propose a way to construct a normalizing flow that *would* have the properties above. More specifically, authors parametrize a $C^k$-continuous univariate diffeomorphism on $[0,1]$, which could then be turned to a diffeomorphism on a circle, torus etc. using methods by Rezende et al. [34], and/or used as the elementwise transformation in coupling/autoregressive layers.

Unfortunately, to make the transformation above expressive, authors have to consider a mixture of simpler transformations, hence the transformation has no analytic inverse. This is not an uncommon limitation when it comes to normalizing flows, where inverting some of the flows (e.g. for sampling) requires performing numerical root-finding. This presents a problem in the context of this work: in literature it has been demonstrated that it is desirable to train the flow on a convex combination of a forward and an *inverse* KL divergences. The latter is the one that presents a problem as it requires sampling from the flow *and* evaluating the gradients of its inverse during training. The cost of doing this can be prohibitive when the inverse is not analytic. Authors derive a way to compute said gradients by *only* using samples from the model and the gradients of the forward transformation, which are typically readily available.

Finally, authors evaluate the proposed solutions on a battery of benchmarks, both synthetic and real, showing that the final model demonstrates desirable behaviour when compared to a strong baseline.


**Limitations And Societal Impact:**

Limitations of the method are discussed by the authors throughout the paper, especially when it comes to the computational costs, though a paragraph in the discussion summarizing said limitations would be useful. The explored applications make it difficult to imagine negative societal impact coming from this work.


**Main Review:**

I should begin by saying that I am not an expert in the domain-specific aspects of the paper, so a few things went over my head. This also means I can't be the judge of the importance of the outlined problems in the context of physical sciences. Having said that, I've been able to follow the motivation behind the work --- authors have done a good job laying out the story behind why the different proposed pieces are necessary. I've enjoyed reading the paper, although it's not the easiest one to wrap your head around.

On the first look the univariate diffeomorphism proposed in section 4 very much resembles the deep sigmoid flow by Huang et al., being a mixture of sigmoid-like functions. Am I correct in understanding that the primary difference is in the proposed transformation being restricted to some interval [a, b] (e.g. [0, 1]), while also being well-behaved/smooth at the boundaries of that interval?

I did not fully understand the reason authors dismiss references [6] and [41] due to their "computational costs". I am not *too* familiar with that literature and the trade-offs of associated methods, but I've seen them applied to large, high-dimensional datasets, so their cost must be at least somewhat manageable? Given that the method authors propose is also "non-analytic", at least in terms of its inverse, I wonder if these references would be worth investigating, especially given their desirable smoothness properties?

The convex combination of the forward and the inverse KL divergences in eq. (5) reminds me of the $\alpha$-divergence: could this be a more principled way to express the desired "distance"?

Section 5 is very cool, especially because the results in eqs. (8)-(11) are immediately applicable to all normalizing flows without an analytic inverse, in cases where it is necessary to evaluate the gradient of the inverse of the flow during training, which I have seen some examples of before.

The experiments are nice: a strong baseline (the rational-quadratic spline) is used, and the differences between it and the proposed method are clear in both the carefully-constructed toy data and real data.

Overall, a well-written paper with a strong motivation, plenty of technical novelty, and robust evaluation. I strongly recommend accepting it to the conference.

---

I thank the authors for their detailed response. I believe the write-up, while already excellent, will benefit from the proposed clarifications. Furthermore, I agree with reviewer *q1W2* that visualisations of individual "ramp" functions, in addition to Figure 1, would be great to aid understanding.

After reading the author's response and the other reviews/responses, I maintain my recommendation for acceptance, although lower my score somewhat to be better calibrated with other reviews, and in recognition of the fact that even though there are multiple useful ideas in this paper, few of them are ground-breaking *by themselves*.

**Time Spent Reviewing:**

8

---

> ### Author Response · Authors · 2021-08-06
> **Answer to Reviewer 9eZb**
>
> Dear reviewer,
>
> we thank you a lot for your time and the encouraging comments. Regarding your remarks, please see our answers below:
>
> > Am I correct in understanding that the primary difference is in the proposed transformation being restricted to some interval [a, b] (e.g. $[0, 1]$), while also being well-behaved/smooth at the boundaries of that interval?
>
> Yes. We will explain this relation more clearly in the final manuscript.
>
> > I did not fully understand the reason authors dismiss references [6] and [41] due to their "computational costs". I am not too familiar with that literature and the trade-offs of associated methods, but I've seen them applied to large, high-dimensional datasets, so their cost must be at least somewhat manageable? Given that the method authors propose is also "non-analytic", at least in terms of its inverse, I wonder if these references would be worth investigating, especially given their desirable smoothness properties?
>
> Reference [6] has a non-analytic forward evaluation that requires approximation techniques (e.g. black box ODE solvers). This can lead to inexact solutions and integration problems, e.g. if the target density is very stiff (as is usually the case for molecular potentials). Another downside of such continuous flow models is the cost of evaluating the Jacobian. Evaluating the exact Jacobian becomes quickly intractable with the dimensionality of the system, whereas noisy estimations, e.g. as done with the Hutchinson estimator, show poor performance in the energy-based training setup and would not provide correct forces. Beyond that, optimizing nODE can become a challenge, as the adjoint method for computing the gradients requires solving an ODE by itself and thus can become numerically difficult. In order to smoothly lift neural ODEs to the circle, recent approaches to nODEs on manifold could be leveraged (see cited references), yet adding another level of complexity.
>
> Reference [41] requires numerical integration of a positive function using some quadrature scheme. This implies a similar problem as mentioned for nODEs: stiff target distribution, such as molecular systems, require high precision and thus many evaluation points. This quickly leads to runtime or memory bottlenecks (depending on implementation). While the inverse of our suggested smooth flows is also obtained with a black-box evaluation method, please consider that forward evaluation is fast and exact. In the case of Ref. [41] each forward evaluation that happens during inversion is again based on numerical integration.
>
> In order to explain the mismatch to the results as obtained on large, high-dimensional datasets: we believe that there can be a difference, whether such models are used for pure density estimation on data, or for approximating a very stiff ground-truth energy as it is in our case. Usually, flows on images are solely evaluated by comparing the negative log-likelihood on holdout data. For our molecular systems this by itself might not help a lot: e.g. bending some torsion angles wrongly might have a minor effect on the likelihood of a sample, yet if it leads to overlapping atoms instantly leads to singular energies. As such, we think that such typical density estimation might have more forgiving characteristics during which these numerical difficulties might not matter as much as we experienced them.
>
> We will write this part more clearly in the final manuscript.
>
> > The convex combination of the forward and the inverse KL divergences in eq. (5) reminds me of the α-divergence: could this be a more principled way to express the desired "distance"?
>
> We chose to use (and modify regarding the force matching error) the original loss taken from Ref. [30] for consistency reasons. As far as we see it, it is not expressable as an alpha-divergence. However, we think looking into alpha-divergences could be a very interesting idea to investigate in future work. At first sight, we do not fully know how to cope with the unknown normalizing constant of the target distribution in this context. We will look into this! Thanks for the hint.

---

### Official Review · Reviewer_QQCj · 2021-07-19

**Rating:** 6
**Confidence:** 3

**Summary:**

This paper is proposing smooth normalizing flows for modeling probability distribution in physical systems.  This is because the existing flow-based model approximates distributions and energies by computing forces and higher-order derivatives based on smoothed energies. This work aims at addressing this challenge by introducing a class of smooth mixture transformations working on both compact intervals and hypertori.  The proposed methods show advantages on two points: 1) enable training by force matching to simulation data and 2) used as potentials in MD simulations.


**Ethics Review Area:**

["I don’t know"]

**Main Review:**

The paper poses an important issue in physical systems and tries to introduce new flow-based approaches to overcome the challenges. Overall, the paper is well-written and clear to follow. However, I have some concerns:

1. The computational cost and scalability would be critical concerns. The authors also mentioned this limitation but I would like to highlight it again. Since the paper aims to solve challenging problems in physical systems, the toy examples or synthesis datasets are not very meaningful. In other words, the impact of the proposed methods would be limited if the algorithm is too computationally intensive for large-scale systems.  The used example in this paper might not be a large-scale example so I can not clearly see the performance in real-world applications.

2. Due to physical motivation, I am not sure how to apply the proposed methods for general problems in the ML community, for example, image datasets. For the baseline methods, have you compared with stochastic NFs [1]? In that paper, some examples may share similar physical motivation and challenges?  Right now, only compared with NSF may not be strong enough to demonstrate the advantages of the proposed smooth NFs.

[1] "Stochastic normalizing flows." arXiv preprint arXiv:2002.06707 (2020).


**Time Spent Reviewing:**

8.5

---

> ### Author Response · Authors · 2021-08-06
> **Anwer to Reviewer QQCj**
>
> Dear reviewer,
>
> we thank you kindly for your time. Please find our answers to your concerns and remarks below:
>
> > The computational cost and scalability would be critical concerns.
>
> We discuss the computational cost of our smooth transformations in the Supplementary Information (Table SI-1) by comparing the non-analytic inversion to the efficient analytic inversion of neural spline flows (NSF). NSFs are known to be computationally cheap compared to most other normalizing flow architectures. The cost of our forward evaluation is comparable to that of neural spline flows. As expected, the non-analytic inversion using standard bijection is more than one order of magnitude slower in both the forward and backward pass. However, by using the proposed multi-bin bijection, the computational cost can be significantly lowered to within a factor of 2-3 of neural spline flows. This relatively minor overhead is easily justified in applications that benefit from the smoothness of the transformations in other ways.
>
> Regarding scalability: the inversion entails only one-dimensional root-finding problems (one for each element of the transformed tensors). Thus, the computational complexity scales linearly in the problem dimension (not considering caching and vectorization effects).
>
> > Since the paper aims to solve challenging problems in physical systems, the toy examples or synthesis datasets are not very meaningful
>
> While alanine dipeptide might be considered a toy example in the context of MD, sampling from its conformational distribution is a challenging task for generative networks. In contrast to the most common generative tasks in computer vision, sampling in physics usually requires a quantitative match of the models’ energies and forces. Even small random perturbations of the atom positions can perturb these energies dramatically, see for instance Fig. 7 and surrounding discussion in our Reference [23]. Due to this highly sensitive, 66-dimensional potential energy surface, a quantitative match with the target distribution is not easily obtained by generative networks. We are not aware of any studies that obtain a similar degree of precision as presented in Fig. 3 when using a neural network for small-molecule conformational sampling.
>
> > The used example in this paper might not be a large-scale example so I can not clearly see the performance in real-world applications.
>
> While applying our smooth flows to larger molecules is a work in progress, the conformational sampling of small molecules is in itself an important scientific problem in many areas, including drug development and chemical engineering. The presented example is a state-of-the-art benchmark and has been used to test neural sampling methods in previous work, see our References.
>
> While running similar experiments on large common benchmark systems like Chignolin or BPTI could be done in principle (here we would refer to Ref. [30]) it would not show anything additional insight relevant for proving our point: that smooth flows are beneficial when trying to match physical potentials.
>
> Running similar tests for such systems is straightforward from a scientific perspective (in the end we would even barely need to change our model), yet it constitutes a significant engineering effort while not providing many new scientific insights for the ML community. It would be an interesting result for scientists from the domain, yet we believe it would be beyond the scope of interests for a broader ML audience.
>
> > Due to physical motivation, I am not sure how to apply the proposed methods for general problems in the ML community, for example, image datasets
>
> The studied problem lies at the core of many problems in molecular physics and is a crucial part of understanding the behavior of proteins and materials.
>
> Also, consider that a method like AlphaFold is inapplicable to image problems, yet clearly constitutes a major contribution to the ML community. On the other hand, innovations in generative models that improve image sampling quality e.g. by introducing useful inductive biases, will likely not benefit molecular sampling.
>
> We see our paper as a meaningful methodological contribution to deriving better inductive biases for flows when applied to the named physical problems. As such, we see it clearly as a machine learning paper that fits the NeurIPS call for papers as “an interdisciplinary conference that brings together researchers in machine learning, computational neuroscience, statistics, optimization, economics, computer vision, natural language processing, computational biology, and other fields” where applications and domain-related work is explicitly encouraged in the call-for-papers.
>
> > For the baseline methods, have you compared with stochastic NFs [1]?
>
> SNFs won’t work with force matching and don’t provide tractable model energies. Running several MD steps between each flow layer also adds tremendously to the computational complexity of the method. Furthermore, they provide an orthogonal contribution that can easily be combined with smooth flows. In the SNF paper, they are used together with affine coupling layers and NSF couplings. So combining them with our smooth flow couplings would be straightforward, yet not providing new insight in this context.
>
> > Right now, only compared with NSF may not be strong enough to demonstrate the advantages of the proposed smooth NFs.
>
> We tried our best to discuss possible alternatives in Sec. 2 and 4 and why we believe they are not suitable in the given situation, e.g. because they are unable to satisfy all requirements simultaneously, or are only able to do that without the unbearable computational overhead and numerical approximation issues (e.g. if pointwise neural ODEs with manifold constraints are used within coupling layers).  If you have any pointers here, we are happy to compare against suitable competing methods that we are unaware of or that are not yet sufficiently discussed in those sections.

---

### Decision · Program_Chairs · 2021-09-27

**Decision:**

Accept (Poster)

**Comment:**

The paper introduces a flow architecture referred to as smooth normalizing flows. These are C^K-smooth maps that work on compact intervals and hypertori.
One of the main contributions is to propose a smooth transformation on the unit-interval.

The paper is overall well-written and technically sound. Conceptually, there is not much innovation in the training setup. But the experimental results are quite promising both with toy and real data. Testing learning models by using them on MD simulations (Langevin diffusions) is a neat idea as it requires that the model log-likelihood's gradient to be sufficiently accurate.

I am quite surprised to see no connections made between "force matching" and the whole literature of "score matching" [e.g. 1, 2, 3, 4, 5, 6] in section 3. I strongly advise the authors to add further discussions and references in this regard.

[1] Hyvärinen, A. and Dayan, P., 2005. Estimation of non-normalized statistical models by score matching. Journal of Machine Learning Research, 6(4).

[2] Song, Y., Sohl-Dickstein, J., Kingma, D.P., Kumar, A., Ermon, S. and Poole, B., 2020. Score-based generative modeling through stochastic differential equations. arXiv preprint arXiv:2011.13456.

[3] Song, Y. and Ermon, S., 2019. Generative modeling by estimating gradients of the data distribution. arXiv preprint arXiv:1907.05600.

[4] Bordes, F., Honari, S. and Vincent, P., 2017. Learning to generate samples from noise through infusion training. arXiv preprint arXiv:1703.06975.

[5] Song, Y. and Ermon, S., 2020. Improved techniques for training score-based generative models. arXiv preprint arXiv:2006.09011.